# Activation of Toll-like receptors nucleates assembly of the MyDDosome signaling hub

Sarah Louise Latty[1], Jiro Sakai[2], Lee Hopkins[2], Brett Verstak[3], Teresa Paramo[4], Nils A Berglund[5], Eugenia Cammarota[6], Pietro Cicuta[6], Nicholas J Gay[3], Peter J Bond[5,7], David Klenerman[1], Clare E Bryant[2]*

[1]Department of Chemistry, University of Cambridge, Cambridge, United Kingdom; [2]Department of Veterinary Medicine, University of Cambridge, Cambridge, United Kingdom; [3]Department of Biochemistry, University of Cambridge, Cambridge, United Kingdom; [4]Department of Biochemistry, University of Oxford, Oxford, United Kingdom; [5]Bioinformatics Institute (A*STAR), Singapore, Singapore; [6]Sector of Biological and Soft Systems, Cavendish Laboratory, University of Cambridge, Cambridge, United Kingdom; [7]Department of Biological Sciences, National University of Singapore, Singapore, Singapore

**Abstract** Infection and tissue damage induces assembly of supramolecular organizing centres (SMOCs)), such as the Toll-like receptor (TLR) MyDDosome, to co-ordinate inflammatory signaling. SMOC assembly is thought to drive digital all-or-none responses, yet TLR activation by diverse microbes induces anything from mild to severe inflammation. Using single-molecule imaging of TLR4-MyDDosome signaling in living macrophages, we find that MyDDosomes assemble within minutes of TLR4 stimulation. TLR4/MD2 activation leads only to formation of TLR4/MD2 heterotetramers, but not oligomers, suggesting a stoichiometric mismatch between activated receptors and MyDDosomes. The strength of TLR4 signalling depends not only on the number and size of MyDDosomes formed but also how quickly these structures assemble. Activated TLR4, therefore, acts transiently nucleating assembly of MyDDosomes, a process that is uncoupled from receptor activation. These data explain how the oncogenic mutation of MyD88 (L265P) assembles MyDDosomes in the absence of receptor activation to cause constitutive activation of pro-survival NF-κB signalling.
DOI: https://doi.org/10.7554/eLife.31377.001

*For correspondence:
ceb27@cam.ac.uk

## Introduction

During infections and tissue injury, large oligomeric complexes of proteins are assembled. The complexes represent supramolecular organizing centers (SMOCs), which serve as the principal subcellular source of signals that promote inflammation (*Kagan et al., 2014*). In the Toll-like receptor (TLR) pathways, the most commonly discussed organizing center is the MyDDosome, which induces NF-κB and AP-1 activation to drive inflammatory transcriptional responses to infection (*Motshwene et al., 2009*; *Lin et al., 2010*; *Gay et al., 2014*). Cell death pathways are also regulated by SMOCs, such as the pyroptosis-inducing inflammasomes and the apoptosis- inducing DISC (*Lu et al., 2014*; *Wang et al., 2010*; *Scott et al., 2009*). A common feature of these organizing centers is their ability to be assembled inducibly during infection or other stressful experiences. Structural analysis has also highlighted similarities in the architecture of SMOCs, in that they assemble into helical oligomers (*Lu et al., 2014*). Currently, it is believed that the purpose of assembling these complexes is to create an activation threshold in the innate immune system, such that all-or-none

**eLife digest** Cells in the immune system have proteins at their surface that detect molecules produced by invading microbes. One of these proteins is Toll-like receptor 4, TLR4 for short. Once TLR4 is activated, the immune cells form MyDDosomes – intricate complexes made of many different proteins. These structures form a signal that mobilizes the cell to fight the infection. In particular, the complexes set up a chain of events that leads to a gene-regulating protein getting access to the cell's DNA. There, the protein switches on genes which produce other proteins important for inflammation, one of the body's most important tools to fight an infection.

The activation of TLR4 is thought to be an all-or-nothing mechanism: the receptors are either 'on' or 'off'. However, different microbial molecules recognized by TLR4 trigger different levels of inflammation, ranging from mild to severe. It remained unclear how an all-or-none response from the frontline receptors could lead to a gradual response from the cell.

Here, Latty et al. compare what happens to TLR4, MyDDosomes and the gene-regulating proteins when living immune cells are stimulated by different doses of two microbial molecules. These agents are both recognized by TLR4, but they lead to different levels of inflammation.

The type of microbial molecule, or their concentration, does not change how TLR4 is activated. Two TLR4 proteins can loosely associate with each together to form a dimer. When they bind a microbial molecule, the dimer becomes more stable. This changes the shape of the TLR4 proteins, which in turn triggers the formation of a scaffold of MyDDosomes. More stable TLR4 dimers are formed when the cells is in contact with a microbial molecule that triggers a strong immune reaction, and possibly when its concentration is higher.

Crucially, the different microbial agents and their concentration levels modify how MyDDosomes assemble. By 'tagging' each protein in the complex with a fluorescent chemical, Latty et al. can follow its formation as it actually happens. When the cells are stimulated with microbial molecules that provoke a strong inflammation, the MyDDosomes may be bigger, in greater numbers, and form more quickly. In turn, under strong microbial activation, the gene-regulating protein that switches on the immune response genes goes to the DNA faster and in higher numbers. This suggests that the pace of assembly, the size and the number of MyDDosomes control the strength of the immune response.

TLR4 is involved in diseases such as cancer or Alzheimer's disease, where the body has an incorrect inflammation response. Knowing in greater detail the cellular processes activated by TLR4 could help efforts to find new drug targets for these conditions.

DOI: https://doi.org/10.7554/eLife.31377.002

responses can be induced during infection. Microbes, however, contain various inflammatory mediators of varying potency. It is unclear how the innate immune system can convert this diversity of microbial stimuli into a digital response, yet single cell analysis of NF-κB activation induced by TLRs suggests that such a response is indeed induced. As TLR-dependent inflammation is controlled by the MyDDosome, an interrogation of this organizing center in living cells may provide an answer to the receptor proximal events that promote inflammation. We considered several possibilities of how MyDDosome assembly can be regulated by microbial ligands of diverse inflammatory potency. First, there may be a direct correlation between MyDDosome size and the inflammatory activity of microbial products. Second, there may be a direct correlation between the number of MyDDosomes assembled and inflammatory activity. Finally, the speed of MyDDosome assembly may dictate the inflammatory activity of individual microbial products. In order to dissect these possibilities, quantitative single cell analysis in living macrophages is required.

## Results

### Rapid assembly and disassembly of MyDDosomes in living cells

We have used single-molecule fluorescence to visualize MyDDosome formation in living cells and analyze the response to lipopolysaccharide (LPS) and a much less active synthetic LPS analogue CRX555. We virally transduced MyD88-GFP into immortalized MyD88$^{-/-}$ macrophages and imaged

the living cells with total internal reflection fluorescence microscopy (TIRFM). MyD88-GFP signaled in both HEK cells and macrophages when over expressed. (*Figure 1—figure supplement 1*; *Figure 1—figure supplement 3*). In unstimulated cells GFP-MyD88 was diffusely distributed at or near the cell membrane, but within 3 min of LPS (500 nM) stimulation macromolecular complexes of MyD88 formed (*Figure 1A*). To estimate the number of MyD88 molecules in these complexes, we compared the intensity of the complexes to those of surface attached dimeric GFP, under identical illumination conditions. The signal intensity of the complexes was approximately three times that of the GFP dimer. This correlates with the 6 MyD88 molecules seen in the crystal structure of the MyDDosome (*Figure 1B*). MyDDosomes can either persist, are rapidly internalized (disappear from the field of view within 30 secs of visualization) or disappear slowly (3 min). The slow disappearance of some MyDDosomes suggests that the MyD88 complex is able to disaggregate presumably to terminate signaling (*Figure 1A*). The formation of 'super' MyDDosomes (2 MyDDosomes coalescing together) was seen in cells stimulated with LPS with more of these complexes formed in response to high concentrations of this ligand (*Figure 1G*). No 'super' MyDDosomes were seen in response to CRX555. These data suggest that the MyDDosome size is likely to be related to signaling efficiency. LPS (500 nM) stimulation also resulted in the rapid formation of many MyDDosome complexes which peaked at 5 min post stimulation within the cell ($p < 0.05$; *Figure 1C*). In response to lower concentrations of LPS (50 nM) or CRX555 fewer MyDDosomes were formed within the same time frame ($p < 0.05$; *Figure 1D*, *Figure 1E*, *Figure 1—figure supplement 4* and *Figure 1—figure supplement 5*).

## TLR4 agonists induce formation of receptor dimers but not higher order oligomers

Current models of signal transduction by TLR4, based on structural analysis, suggest that the TLR4/MD2 co-receptor assembles into a hetero-tetramer when bound to immunostimulatory LPS and that this induces dimerization of the cytosolic Toll/interleukin-1 receptor (TIR) domains. This complex then acts as a scaffold for the recruitment of downstream signal transducers Mal/TIRAP and MyD88 to form a membrane-associated signalosome. Experimental evidence to support this model, however, is lacking. To investigate this question we developed a single-molecule fluorescence approach to quantify TLR4 dimerization on the cell surface in response to LPS and CRX555. A HaloTag was added to the C-terminal of TLR4 and the construct transduced into immortalized TLR4$^{-/-}$ macrophages (iBMMs). HaloTagged TLR4 signaled in response to LPS in HEK cell reporter assays (*Figure 2—figure supplement 1*). Cells were incubated with HaloTag R110Direct for 30 min to label the HaloTag and, following three wash steps, incubated with or without ligand, placed on glass cover slides and fixed at different time points. The cell membrane was imaged using TIRFM and subject to photobleaching analysis to determine the oligomerization state of the labeled TLR4 molecules present. Labeled monomers of TLR4 photobleach in a single step whilst labeled dimers will photobleach in two steps (*Figure 2A(ii)(iii)*). Due to the absence of a good antibody to TLR4 we were unable to determine the efficiency of labeling of TLR4, so our measurements may underestimate the dimer population but allow us to follow relative changes in the number of TLR4 monomer and dimers. Unstimulated cells showed two different populations of TLR4 complexes present: monomeric (78 ± 3%) and dimeric (22 ± 3%) (*Figure 2B(i)*), indicating the labeling level of TLR4 must be at least greater than 30%. The number of TLR4 dimers on the cell surface is small and stimulating cells with LPS showed an increased number of dimers at 5 min following stimulation, compared to unstimulated cells, which rapidly reduced presumably due to internalization of TLR4 and trafficking to the endosome (*Kagan et al., 2008*) (*Figure 2B(ii)*). This trend was clearest at lower levels of LPS because at higher concentrations TLR4 internalization occurred very rapidly. At very low concentrations of LPS or with CRX555 the number of dimers formed was comparable to that seen in unstimulated cells (*Figure 2B(iii)*). We did not observe clusters of TLR4 with any ligand and thus higher order oligomerization of the receptors is unlikely.

In contrast to stimulatory LPS, structurally related antagonists such as Eritoran or *Rhodobacter sphaeroides* Lipid A (RSLA) bind to TLR4/MD2 co-receptors but do not induce the formation of hetero-tetrameric complexes in vitro. Consistent with this we find that the number of TLR4 dimers on the surface of macrophages treated with RSLA is significantly lower than that seen in unstimulated cells (*Figure 2B(iv)*). This suggests that RSLA stabilizes the TLR4/MD2 heterodimer and prevents the formation of constitutive tetramers. We next used a TIR domain mutant of TLR4, Pro712His, that is unable to signal (*Poltorak et al., 1998*) or recruit downstream signal transducers such as MyD88 to

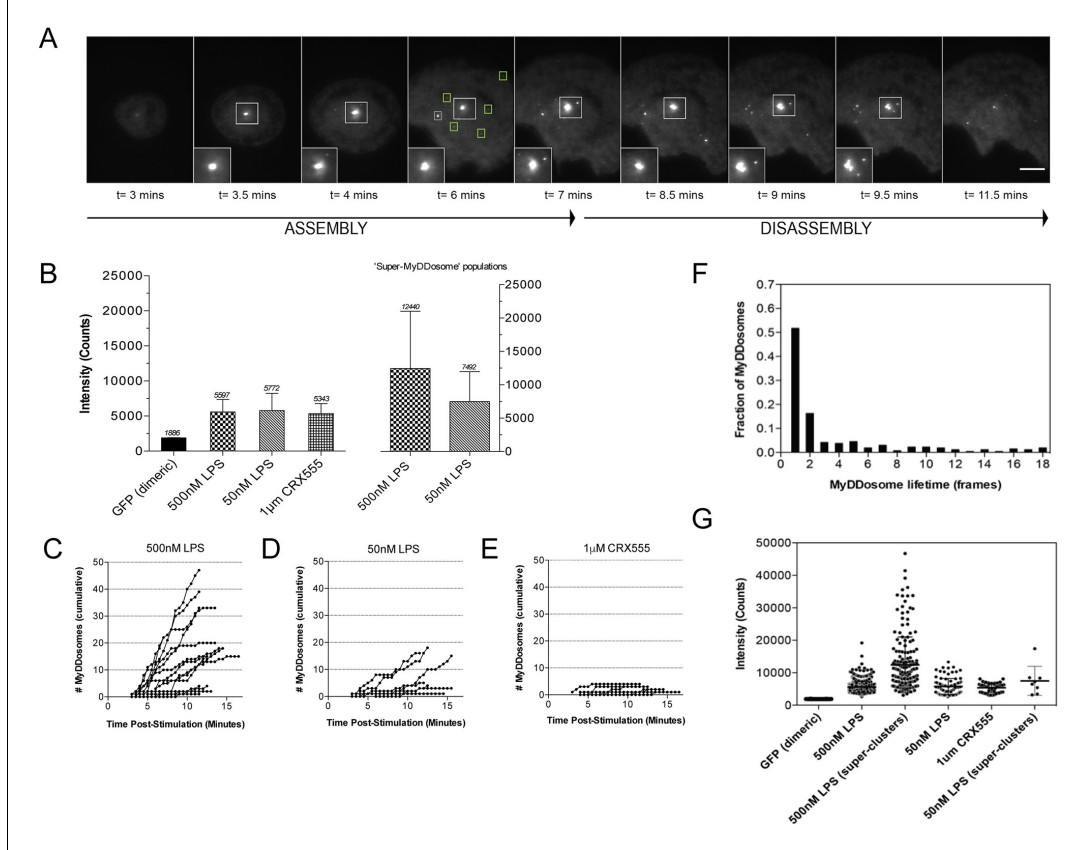

**Figure 1.** The kinetics and size of MyDDosome formation depend on the efficacy of TLR4 stimulation. MyD88$^{-/-}$ iBMM were virally transduced with pHRMyD88-GFP and either left unstimulated or stimulated with LPS (10–500 nM) or CRX-555 (500 nM-1μM). (**A**) Stills showing MyDDosome formation in a live cell in response to 500 nM LPS stimulation. Imaging was started 3 min post-stimulation and the assembly of MyDDosomes was observed up until 8 min following stimulation. After 8 min the MyD88 complex can disassociate. Scale bar is 5 μm. (**B**) Mean fluorescent intensities of MyDDosomes post-stimulation. Mean intensity values are displayed above the error bars. These values are compared to the distribution obtained from dimeric recombinant GFP. Error bars represent SEM. (**C, D, E**) The cumulative number of MyDDosomes per cell (each trace represents a single cell) for 500 nM LPS, 50 nM LPS and 1 μM CRX555 stimulations respectively. Single MyDDosomes are surrounded by green boxes. In response to a high dose of LPS (500 nM) a large MyDDosome population of varying sizes forms. Upon stimulation with lower concentrations of LPS (50 nM) fewer MyDDosomes were formed within the same time frame (p<0.05). The partial agonist CRX555 (1 μM) results in less cumulative MyDDosome formation than LPS (500 nM; p<0.0001). A minimum of 10 cells were analyzed for at least three repeats per condition. (**F**) Histogram of MyDDosome lifetime (frames) following stimulation with 500 nM LPS. 1 frame = a 30 s interval. Many tracks vanish in one frame but a proportion are slower. A minimum of 10 cells were analyzed for at least three repeats per condition. (**G**) Comparison of the fluorescence intensity of dimeric GFP with MyD88 in response to stimulation with different TLR4 ligands. Larger clusters of MyD88 were seen in response to LPS, particularly at higher doses. Data are presented with error bars as standard deviation of the intensity values.

DOI: https://doi.org/10.7554/eLife.31377.003

The following figure supplements are available for figure 1:

**Figure supplement 1.** (i) To determine whether addition of a fluorescent tag to MyD88 effects its ability to signal HEK cells were transiently transfected with 10 ng MyD88-GFP, along with an 10 ng p-NFκB Luc reporter and 5 ng of phRG (constitutively active renilla control plasmid).
DOI: https://doi.org/10.7554/eLife.31377.004

**Figure supplement 2.** An overview of the MyDDosome tracking process using a TrackMate plugin.
DOI: https://doi.org/10.7554/eLife.31377.005

**Figure supplement 3.** Representative MyD88-GFP data for 500 nM LPS treated cells.
DOI: https://doi.org/10.7554/eLife.31377.006

**Figure supplement 4.** Representative MyD88-GFP data for 50 nM LPS treated cells.
DOI: https://doi.org/10.7554/eLife.31377.007

**Figure supplement 5.** Representative MyD88-GFP data for 1 μM CRX-555-treated cells.
DOI: https://doi.org/10.7554/eLife.31377.008

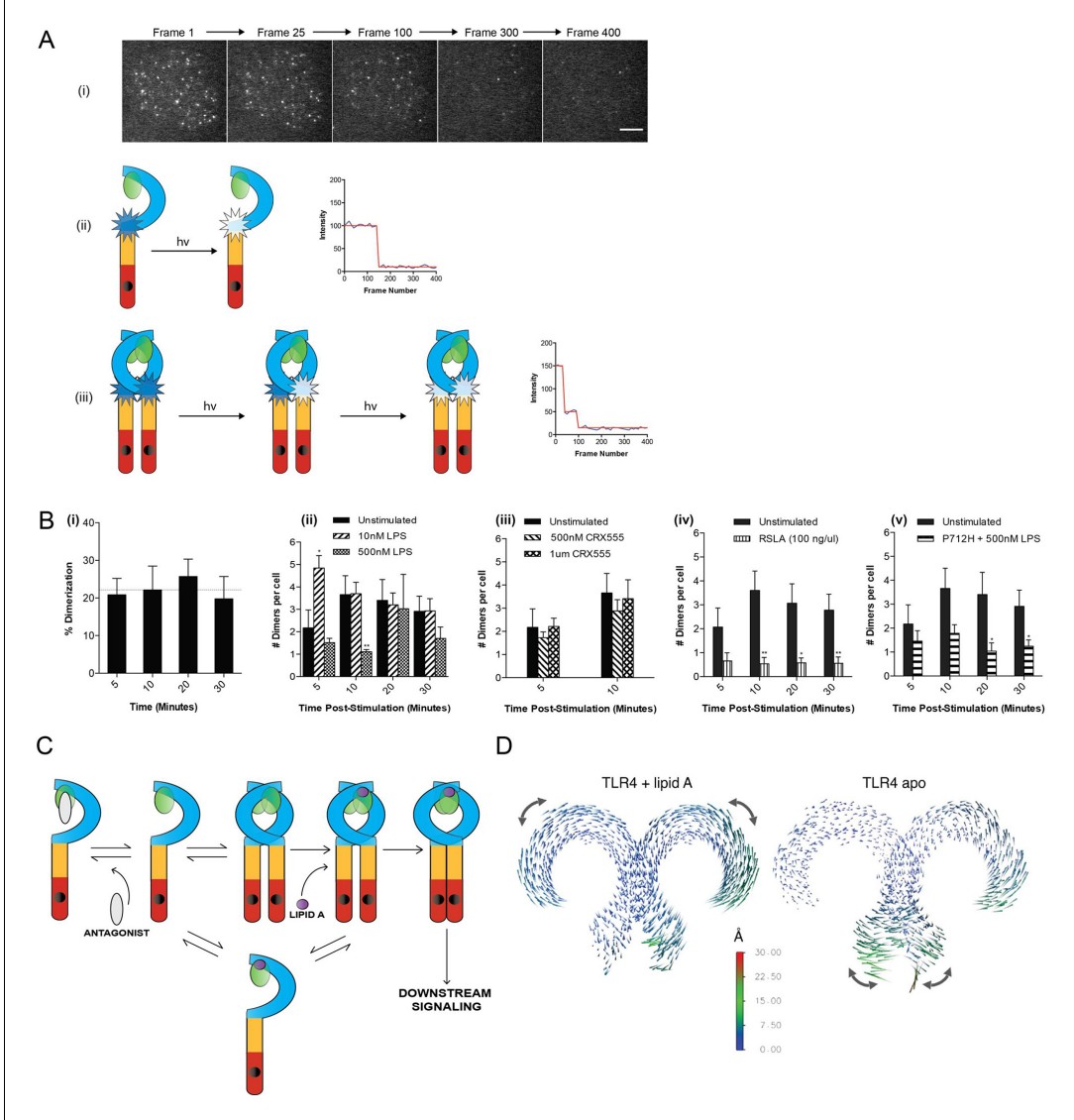

**Figure 2.** The effect of ligands upon TLR4 dimerization. TLR4$^{-/-}$ iBMM virally transduced with pHR-TLR4-Halo were incubated with HaloTag R110Direct. Cells were either left unstimulated or stimulated with LPS (10–500 nM) or CRX-555 (500 nM-1μM), fixed at specific time points and subjected to photobleaching analysis. (**A**) Photobleaching analysis with stills from cells stimulated with LPS (10 nM) for 5 min. If the labeled TLR4 is a monomer then photobleaching occurs in a single step. If the labeled TLR4 is a dimer then two photobleaching steps occur. Since it was not possible to independently determine the efficiency of labeling of TLR4, this analysis allows the relative number of TLR4 monomers and dimers on the cell surface to be determined. (**B**) (i) TLR4 expression in unstimulated cells: monomeric (78 ± 3%); dimeric (22 ± 3%) (ii) increased number of detected TLR4 dimers after LPS stimulation (10 nM) for 5 min ($p < 0.05$) (iii) the number of detected TLR4 dimers is unchanged by CRX555 (iv) less TLR4 dimers were detected with the antagonist *Rhodobacter sphaeroides* Lipid A (RSLA) compared to unstimulated cells ($p < 0.05$) and (v) LPS-stimulation of TLR4-Pro712His-Halo shows reduced number of dimers compared to wild-type TLR4 ($p < 0.05$). At least 16 cells were analyzed at each time point in three independent repeat experiments, data are expressed as mean ± SEM, and data were analyzed by a two-tailed unpaired Student's t-test. (**C**) A two-step model for TLR4 signaling: ligand induced dimer stabilization followed by apposition of the TIRs. (**D**) Porcupine plots from molecular dynamics simulations of TLR4/MD2, with magnitudes of atomic motion indicated by length and color of associated arrows: minor rotational motions of the ECDs with lipid A brings the C-termini of the TLR4 ECD into close apposition.

DOI: https://doi.org/10.7554/eLife.31377.009

The following figure supplements are available for figure 2:

**Figure supplement 1.** (i) To determine whether addition of a Halo tag to TLR4 effects its ability to signal HEK cells were transfected with 1 ng Wild-type (TLR4WT) or Halo-Tagged TLR4 (TLR4Ha), 1 ng each of CD14 and MD2, 10 ng p-NFκB Luc reporter and 5 ng of phRG (constitutively active renilla control plasmid).

DOI: https://doi.org/10.7554/eLife.31377.010

*Figure 2 continued on next page*

*Figure 2 continued*

**Figure supplement 2.** Porcupine plots based on three independent replica simulations of apo, ligand-free TLR4/MD2, with magnitudes of atomic motion indicated by length and color of associated arrows, reveal large lateral fluctuations of C-termini, consistent across all replicas.

DOI: https://doi.org/10.7554/eLife.31377.011

**Figure supplement 3.** Dynamic motion of MD2 relative to TLR4.

DOI: https://doi.org/10.7554/eLife.31377.012

ask whether assembly of the TLR4/MD2 tetramer occurred in the absence of signal transduction. We compared the numbers of TLR4 monomers and dimers in unstimulated and LPS-stimulated cells transduced with HaloTagged TLR4 Pro712His to the numbers seen for wild-type TLR4. The ratio of monomer to dimer Pro712His TLR4 was similar to wild type TLR4 in unstimulated cells, but in LPS-stimulated cells it was less than that seen for cells transduced with native HaloTagged TLR4 (*Figure 2B(v)*). This result indicates that LPS binding to the Pro712His receptor stabilises the TLR4/MD-2 heterodimer and inhibits the formation of active tetramers. Dimerisation of the TLR4/MD-2 ectodomain and the TIR domain BB-loop are thus synergistic and both events must occur to produce an active receptor complex. This implies a two step activation mechanism for TLR4, as illustrated in *Figure 2C*. Alternatively, failure of the P712H receptor to recruit adaptors could lead to rapid removal of the inactive receptor heterotetramers from the cell surface.

## A two-step mechanism for TLR4 activation

Dynamic molecular modeling has suggested that binding of different ligands causes conformational changes in MD2 that may be linked to receptor function (*Paramo et al., 2013*). We analyzed the final relative location of MD2 in the dimeric TLR4 complex by dynamic molecular modeling and discovered that the Lipid A agonist-associated state remained close to the LPS-bound X-ray structure (*Song and Lee, 2012*), whereas in the absence of agonist, a shift of up to ~10 Å of MD2 relative to its primary TLR4 partner was observed as it disassembled from the secondary, dimeric TLR4 interface. The conformation of the two TLR4/MD2 heterodimers observed in this modeled structure (*Figure 2D*, right hand) closely resembled the X-ray structures of monomeric TLR4 in the unliganded-mouse and Eritoran antagonist-bound human constructs. (*Song and Lee, 2012*)

Principal component analysis (PCA) was then performed on the dynamic trajectories in order to isolate the dominant, collective motions of the dimeric TLR4 chains (*Figure 2D*, *Figure 2—figure supplement 2*). This revealed that in the presence of Lipid A there were minor rotational motions of the ECDs with respect to one another at equilibrium, similar to the 'ring rotation' observed with ligand bound TLR8, but no significant separation of the two C-termini (*Ohto et al., 2014*) (*Figure 2—figure supplement 2*, *Figure 2—figure supplement 3*). In the absence of Lipid A, large lateral fluctuations of the C-termini were observed, similar to the 'hinge motion' seen in the TLR8 inactivated homodimer (*Ohto et al., 2014*). Consistent with these observations, in the absence of ligand, the backbone of the heterotetrameric TLR4/MD2 complex was increased by ~50–60% whilst up to 60% of the surface area buried between each TLR4/MD2 heterodimer and its adjacent primary TLR4 partner was lost, compared to the lipid A bound state (*Supplementary file 1*), indicative of major conformational changes. Collectively, these data provide further support for a two-step TLR4 activation model where ligand binding results in a structural change to facilitate close apposition of the C-termini of the TLR4 dimer to allow TIR dimerization.

## Signal strength is determined by MyDDosome assembly kinetics

To address further the coupling of receptor activation and signal transduction, macrophages were stably transduced with the NF-κB subunit RelA tagged with GFP and a TNFα-reporter construct fused to mCherry (*Sung et al., 2014*) and were stimulated with increasing concentrations of the TLR4 agonist LPS or the partial agonist CRX555 (*Stöver et al., 2004*). Single cells were visualized using live confocal microscopy for 24 hr (*Figure 3A*). The dynamics of RelA-GFP translocation into the cell nucleus and TNFα-mCherry induction were quantified for individual cells (*Figure 3—figure supplement 1*). LPS induced rapid translocation of NF-κB into the nucleus and the rate of nuclear translocation increased with the dose of LPS (*Figure 3B*). The fastest time observed from initial stimulation to peak levels of nuclear NFκB was 20 min, findings that correlate well with electrophoretic mobility shift assays where accumulation of NFκB is seen within minutes after cellular stimulation.

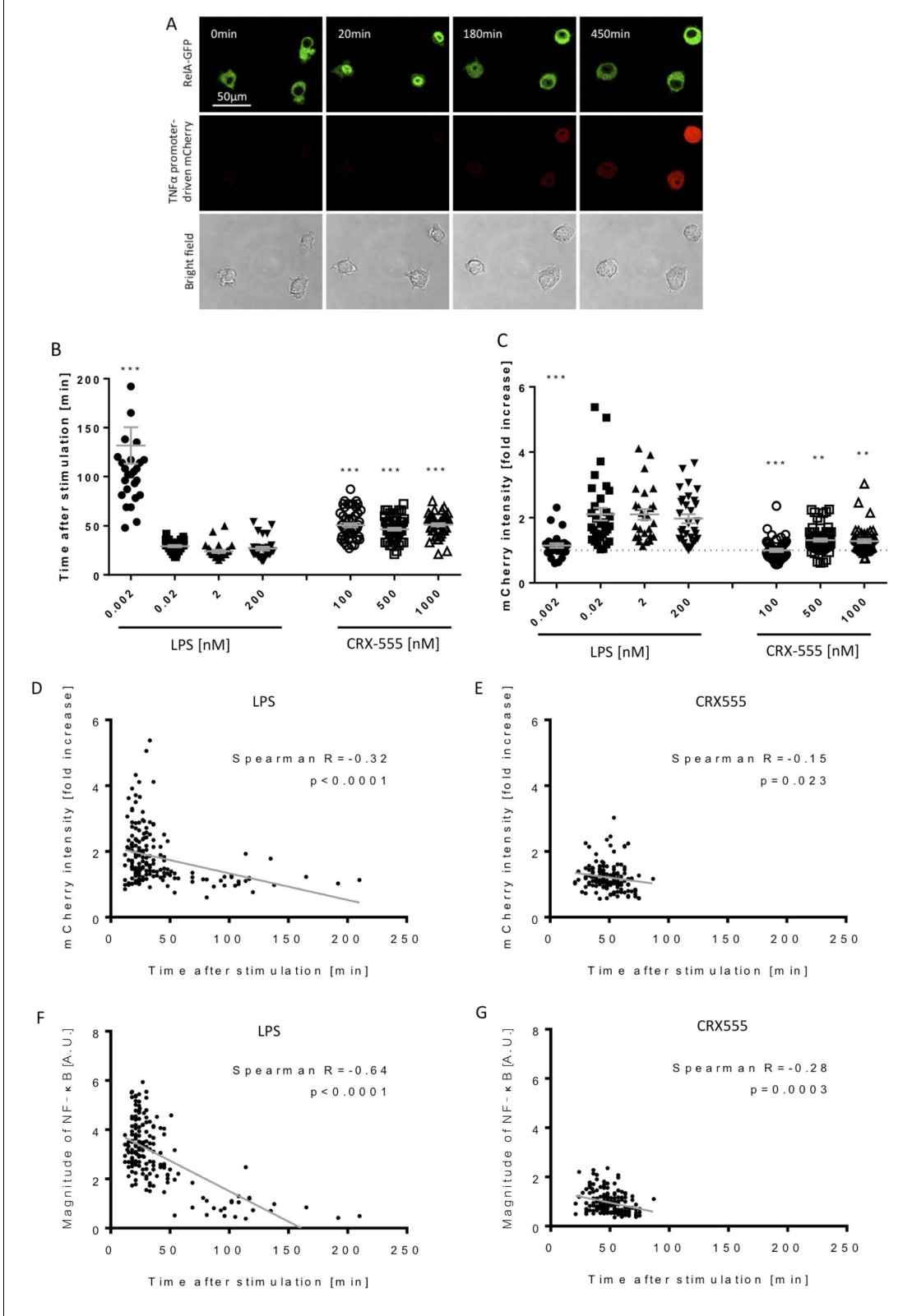

**Figure 3.** Macrophages stimulated with CRX555 show delayed NF-κB nuclear translocation in comparison to cells stimulated with LPS. RAW264.7 macrophages stably expressing RelA-EGFP and a TNFα promoter-mCherry reporter were stimulated with LPS or CRX555. Confocal time-lapse images were captured every 3 min for 15 hr. (**A**) Stills of NF-κB nuclear translocation and mCherry expression following 2 nM LPS stimulation. (**B**) NF-κB dynamics was assessed as the ratio of the nuclear:cytoplasmic EGFP fluorescence. Time to the first peak of NF-κB translocation was measured in

*Figure 3 continued on next page*

*Figure 3 continued*

response to LPS or CRX-555. (C) Fold increase of mCherry intensity following stimulation with LPS or CRX555 (the dotted line indicates no increase in mCherry). Each dot represents a single cell from four independent experiments (n = 30–50 for each condition). The bar represents mean value ± s.d. ***p<0.0001, **p<0.01, *p<0.05, one-way ANOVA with Kruskal- Wallis post-test, compared with cells stimulated with the lowest concentration of LPS (0.002 nM). (D and E) Statistical analysis of the correlation between the speed of NF-κB nuclear translocation and TNFα promoter activation. Spearman's correlation coefficient was used to assess links between time to the first peak of NF-κB nuclear translocation and the fold increase in mCherry intensity or magnitude of NF-κB translocation following LPS (D) (F) or CRX-555 stimulation (E) (G).

DOI: https://doi.org/10.7554/eLife.31377.013

The following figure supplement is available for figure 3:

**Figure supplement 1.** MATLAB-based automated analysis of NF-κB translocation and mCherry fluorescence intensity.

DOI: https://doi.org/10.7554/eLife.31377.014

(*Kopp and Ghosh, 1995*) Low concentrations of LPS or stimulation of cells with CRX555 showed delayed translocation of NF-κB into the nucleus coupled to reduced expression of the TNFα-mCherry reporter construct (*Figure 3B and C*). Statistical analysis of the individual parameters of the single cell signal transduction assays shows that TNFα-mCherry production was significantly correlated to the speed of NF-κB translocation into the nucleus (*Figure 3D and E*). The peak timing of NFκB translocation also correlates with the magnitude of NFκB translocation and the TNF α-mCherry reporter expression for both LPS and CRX555 stimulated cells. These kinetic data correlated with the kinetics of our single cell signaling assays suggesting that the strength of signal is also partially determined by the rapidity with which a critical number of MyDDosomes form to trigger NF-κB translocation to the nucleus.

## Discussion

Here we show that in the absence of ligand, TLR4/MD2 on the surface of living immune system cells is in a dynamic equilibrium with populations of heterodimers and heterotetramers. The binding of LPS likely stabilizes the tetrameric form and initiates conformational changes that lead to signal transduction. In the absence of ligand, the ECDs of the endosomal TLR8 exist as stable, inactive preformed dimers. The binding of small molecule agonists causes an extensive conformational rearrangement within the dimer that brings the C-termini of the TLR8 ECDs closer together (*Tanji et al., 2013*). Our molecular dynamic analysis suggests that TLR4 activation involves a similar process in which the two ECDs tilt and rotate with respect to each other during signal transduction. This also suggests that the crystal structure of heterotetrameric TLR4/MD2 might represent an inactive transition state corresponding to the first step of the concerted activation process (Figure 2C, [*Gay et al., 2006*])

We also visualize, for the first time, the assembly of the membrane associated MyDDosome signaling scaffold in vivo and show that signaling flux depends on the size and number of these structures. The absence of oligomeric receptor clusters also implies that active TLR4 does not form a stoichiometric post-receptor complex with the MyDDosome. A dimer of the receptor TIR domains is assumed to have two binding sites for MyD88 (*Valkov et al., 2011*; *Núñez Miguel et al., 2007*) whereas the Myddosome has about 6 MyD88 molecules both in vitro and in vivo. This stoichiometric mismatch taken together with the highly transient nature of the Myddosome (*Figure 1*) indicates that the activated receptor nucleates the assembly of the higher order MyDDosome structures that associate only transiently with the membrane bound receptor rather than forming a stable signalosome (*Motshwene et al., 2009*; *Triantafilou et al., 2004*). This assembly mechanism is similar to that proposed for the pyrin domain of Asc which nucleates assembly of filamentous NLRP3 inflammasomes (*Lu et al., 2014*). Rapid dissociation of the MyDDosome from the receptor is likely to be coupled to TLR4/MD-2 internalisation providing a mechanism for the sequential activation of the MyD88 and TRIF directed signals from the cell surface and endosomes respectively (*Kagan et al., 2008*; *Zanoni et al., 2011*). These findings also explain the properties of an oncogenic somatic mutation in MyD88 (L265P) commonly found in B-cell lymphomas and other conditions such as Waldenstrom's macroglobulinemia (*Treon et al., 2012*; *Ngo et al., 2011*). In these diseases, MyDDosomes assemble spontaneously in the absence of receptor activation causing constitutive activation of NFκB which acts as a pro-survival signal (*Avbelj et al., 2014*). Our results are consistent with

another model of cooperative assembly for the Mal/TIRAP adaptor that was proposed recently *Ve et al., 2017*

Our data show that partial agonists stimulate the formation of a smaller number of MyDDosomes more slowly than the agonist LPS and that this leads to slower translocation of NF-κB into the nucleus. LPS forms a larger number of MyDDosomes more rapidly resulting in more rapid NF-κB translocation. Surprisingly only a small number of MyDDosomes need to be formed on the cell surface for full cellular signalling to occur and we proposed that the difference between full and partial agonism is determined by MyDDosome number and the speed of their formation. This can simply be controlled by the equilibrium between the agonist or partial agonist for the TLR4 dimers present on the cell surface, with agonists having faster on-rates or slower off rates than the partial agonists that is a higher affinity. This provides a simple explanation of how a graded response is achieved in TLR4 signalling.

The small number of MyDDosomes needed for signalling is reminiscent of the situation with the T-cell recptor (TCR). In this case 10 TCRs can activate a T-cell with each triggered TCR rapidly forming a larger signalling complex (*Smith-Garvin et al., 2009*; *Irvine et al., 2002*). Both TCR and TLR4 signalling are highly sensitive, show graded responses and occur at the level of the single molecule requiring a mechanism to be in place to prevent inadvertent signalling. For the TCR this is based on the affinity of the ligand presented by the MHC to the receptor, and this provides for discrimination between self and non-self by mechanisms that still need to be elucidated (*Huang et al., 2013*). For TLR4 it seems that receptor dimers need to be stabilised by an agonist in order to signal. Affinity is also the mechanism used by TLR4 to obtain a graded response by stabilising TLR4 dimers to greater or lesser extents. In both TCR and TLR4 signalling a larger signalling complex is formed as a result of a single molecule event leading to significant signal amplification.

In summary, our study suggests that TLR4 signalling occurs at the level of single molecules with agonists stabilising low numbers of preformed dimers that then nucleate the formation of a short-lived MyDDosome signalling complex which is then removed from the cell surface. Partial agonists form less MyDDosomes more slowly leading to a smaller overall cellular response. This provides new insights into the mechanism of TLR4 signalling and how it is possible to obtain a graded response to different agonists.

## Materials and methods

### Single cell signaling assays

The RAW264.7-derived reporter cell line which expresses enhanced green fluorescent protein (EGFP)-tagged RelA and TNFα promoter-driven mCherry was provided by Dr. Iain D.C. Fraser (National Institute of Health, MD, USA). The cells were maintained in Dulbecco's Modified Eagle's Medium (DMEM; Sigma-Aldrich) supplemented with 10% (v/v) heat-inactivated fetal calf serum (FCS; Thermo Scientific, Rugby, UK), 2 mM L-glutamine (Sigma-Aldrich) and 20 mM HEPES (Sigma-Aldrich) at 37°C, 5% $CO_2$. The cells were plated on a 35 mm glass-bottom dish (Greiner Bio-One) at a concentration of $1.0 \times 10^5$ cells in phenol red-free DMEM supplemented with 10% (v/v) FCS, 2 mM L-glutamine and 20 mM HEPES, and settled down in an incubator for 8 hr prior to experiment. The dish was mounted on the stage of a confocal microscope (TCS SP5, Leica) and kept at 37°C, 5% $CO_2$ during the experiment in a climate chamber. Live cell imaging was performed immediately after stimulating the cells. Images were sequentially taken on a 40x oil-immersion objective (NA1.25) with 2.0x zoom every 3 min for 15 hr. The image dimension used was 512 × 512 pixels. The pinhole size was 1 A.U., and the thickness of the focal plane was 0.96 μm. Acquired images were exported as 16-bit TIFF files for analysis. A MATLAB-based automated single cell analysis script was used to automatically assess NF-κB nuclear translocation and TNFα promoter-driven mCherry expression (*Figure 3—figure supplement 1*).

### Ligand Preparation

Lipopolysaccharide (LPS) and *Rhoderbacter sphaeroides* lipid A (RSLA) stocks were thawed from storage at −20°C and sonicated for 1 min. The ligand was then diluted to the appropriate concentration in DMEM. CRX555 stored at 4°C was sonicated for 1 min prior to dilution to the required concentration with DMEM.

## Single-molecule fluorescence analysis

### Plasmids and constructs

For transient transfection assays pCMV-TLR4, pEFIRES-MD-2, pCMV-CD14, pNF-κB-luc (Clontech) and phRG-TK (Promega) were used. Construct design and molecular cloning: DNA encoding full-length hTLR4 from pCMV8-Flag-hTLR4 vector and hMyD88 from pBK-CMV-Myc-MyD88 were subcloned with MluI and BamHI sites into lentiviral plasmid pHR'CMVlacZ-Halo, a self-inactivating HIV-1 vector with a C-terminal Halo-Tag. Plasmid pHR'CMVlacZ-MyD88-GFP was generated by subcloning the GFP gene from commercially sourced vector pEGFP-N1 (Clontech) with BamHI and NotI sites for ligation into pre-digested pHR'CMVlacZ-MyD88-Halo at the corresponding sites.

### Cell culture, transfection and viral transduction

HEK 293T and iBMMs (TLR4$^{-/-}$ (NR-9458) and MyD88$^{-/-}$ (NR-15633) were a gift from Doug Golenbock and Kate Fitzgerald now banked with BEI Resources, USA) were maintained in DMEM-Complete medium (DMEM, 10% FCS, 2 mM L-glutamine and 100 U/ml penicillin/100 µg/ml streptomycin) at 37°C, 5% CO$_2$. All cell lines were characterised to ensure the mouse gene of interest was absent, that the cellular response to LPS was altered as expected and that the cells retained a macrophage-like phenotype. All cell lines are routinely tested to ensure they remain mycoplasma free. To determine whether addition of a tag to TLR4 and MyD88 would prevent these constructs from signalling HEK cells were seeded into a 96 well flat bottomed plates at a density of $3 \times 10^4$ cells/well 48 hr prior to transfection. Cells were transfected with wild type or tagged TLR4 (1 ng), MD-2 (1 ng), CD14 (1 ng) or tagged MyD88 (10 ng) in conjunction with the reporter vectors pNF-κB-luc (10 ng) and phRG-TK (5 ng) per well using jetPEI (Polyplus) according to the manufacturer's instructions. Cells transfected with TLR4/MD2/CD14 were stimulated with 1 or 10 ng/ml ultrapure LPS (Invivogen) 48 hr post transfection. Cells transfected with MyD88 were lysed 24 hr after transfection.

For virus production HEK 293T seeded at approximately 50% confluency in 12-well tissue culture plates were transfected using a 3:1(µl:µg) ratio of Genejuice (Novagen) to DNA according to the manufacturer's instructions. A total of 1.5 µg DNA (500 ng each of p891, pMDG and pHR TLR4Halo or pHR MyD88GFP) was transfected per well. Cell culture supernatants containing lentiviral particles were harvested four days post transfection and centrifuged for 5 min at 1000 rpm. Neat or diluted clarified supernatants were added to the appropriate target knockout murine macrophage cell line (TLR4$^{-/-}$ or MyD88$^{-/-}$) and incubated for 24 hr. Supernatants were discarded and replaced with DMEM-Complete and transduced cells were incubated for an additional 24 hr prior to harvesting and subsequent analysis.

HEK 293T and iBMMs (TLR4$^{-/-}$ and MyD88$^{-/-}$) were maintained in DMEM-Complete medium (DMEM, 10% FCS, 2 mM L-glutamine and 100 U/ml penicillin/100 µg/ml streptomycin) at 37°C, 5% CO$_2$. HEK 293T seeded at approximately 50% confluency in 12-well tissue culture plates were transfected using a 3:1 (µl:µg) ratio of Genejuice (Novagen) to DNA according to the manufacturer's instructions. A total of 1.5 µg DNA (500 ng each of p891, pMDG and pHR TLR4$^{Halo}$ or pHR MyD88$^{GFP}$) was transfected per well. Cell culture supernatants harboring lentiviral particles were harvested four days post transfection and centrifuged for 5 min at 1000 rpm. Neat or diluted clarified supernatants were added to the appropriate target knockout murine macrophage cell line (TLR4$^{-/-}$ or MyD88$^{-/-}$) and incubated for 24 hr. Supernatants were discarded and replaced with DMEM-Complete and transduced cells were incubated for an additional 24 hr prior to harvestation and subsequent analysis.

### TLR4 Halo-Tag labeling

A stock solution of HaloTag R110 Direct Ligand (Promega) was diluted in DMEM to a final concentration of 100 nM. All medium was removed from the cells and replaced with the DMEM containing 100 nM of Halo-Tag R110Direct Ligand (Promega) before incubation for 30 min at 37°C. Cells were then washed with $2 \times 1$ ml DMEM and incubated for a further 60 min in 500 µl of DMEM. A final wash with $1 \times 1$ ml DMEM preceded an additional incubation period of 30 min in 500 µl of DMEM. In single cell imaging we observe approximately 30–50 TLR4 tracks per cell, over an average area of 100 µM$^2$. Assuming 33% efficiency in labelling of TLR4-Halo (*Latty et al., 2015*) then the estimated number of TLR4 molecules is approximately one per µM (*Motshwene et al., 2009*). This is similar to

the levels of TLR4 described when transfected into HEKs at levels considered comparable to endogenous TLR4 in glioma cells (*Krüger et al., 2017*).

## Sample Preparation for Microscopy

Microscope coverslips were plasma cleaned (Harrick Plasma, PDC-002) in an Argon atmosphere for 60 min before subsequent coating with Polylysine-grafted Polyethyleneglycol (PLL-g-PEG, SuSoS) for 45 min. Slides were then washed in duplicate with filtered (0.22 µm Millex-GP syringe filter unit) phosphate buffered saline (PBS, Life Technologies). Following labeling, TLR4-Halo transfected cells were resuspended in supplemented DMEM (DMEM supplemented with 10% FCS, 1% Antibiotics and 1% L-Glutamine), mechanically removed and centrifuged at 2000 rpm for 90 s. The resulting supernatant was removed and cells were resuspended in 200 µl DMEM (with or without ligand) before being added to the coated cover slides. There is an inherent dead time of a few minutes due to the time taken for the cells to adhere to the cover slides preceding imaging. Alignment of the instrument for TIRF imaging also prevents us from probing earlier time points. The plated cover slides were moved to 37°C and allowed to settle for the desired time (5, 10, 20, 30 min) prior to fixation. To fix the cells, medium on cover slides was replaced with 4% formaldehyde solution (16% w/v stock solution, Thermo Scientific, diluted to 4% with PBS). The formaldehyde solution was left on the plated cells for 1 hr at constant room temperature before replacement with PBS immediately preceding imaging. Reconstituted MyD88$^{-/-}$ macrophages with GFP labeled MyD88 were resuspended in supplemented DMEM (DMEM supplemented with 10% FCS, 1% Antibiotics and 1% L-Glutamine), mechanically removed and centrifuged at 2000 rpm for 90 s. The resulting supernatant was removed and cells were resuspended in 200 µl DMEM +2% HEPES buffer (Sigma) (with or without ligand) before being added to the coated cover slides on the microscope stage at 37°C.

## Total internal Reflection Microscopy (TIRFM) experimental set-up

TIRFM was utilised to image the prepared samples. The fluorescence signal detected is within ~100 nm of the basal surface of the cell due to limitation imposed by the evanescent wave range. A compact solid-state frequency-doubled laser operating at 488 nm (Cyan Scientific, Spectra Physics) was utilised for TIRFM. The 488 nm beam enters the microscope on the edge of the back focal plane of a 1.45 NA TIRF objective (60 x Plan Apo TIRF, NA 1.45, Nikon) mounted on a Nikon TE2000-U microscope. A dichroic (490575DBDR, Omega Optical) separated the collected fluorescence from the returning TIR beam. The fluorescence component was then split into red and yellow components (585 DXLR, Omega Optical) and filtered using Dual-ViewTM (Optical Insights) mounted filters. Images were acquired on an EMCCD equipped with a dual view imaging system (Cascade II + DV2: 512 Princeton Instruments), the EMCCD split such that the 488 fluorescence was visible on one half of the device (−70°C; dichroic: DV2 FF562-Di03, Semrock). Data acquisition of TLR4 photobleaching data was performed using Micromanager Software, Version 1.4.13. Image stacks were recorded over 400 frames with exposure set at 35 ms. The operating power density for the 488 nm laser used to acquire data sets was 5.19Wcm$^{-2}$. MyD88 data sets were acquired as image stacks with one image taken every 30 s to collect 18 frames of data with exposure set at 100 ms. Acquisition over the 9 min time period took place at 37°C (using a stage incubator with enclosure (digital pixel imaging systems)). The operating power density for the 488 nm laser used to acquire data sets was 5.19Wcm$^{-2}$.

## Photobleaching Analysis

Spots were identified from image stacks using a previously published custom tracking algorithm (*Weimann et al., 2013*). Spots were localized in each frame of the input TIFF file following calculation of their respective centroid positions. Following spot localization, spots were connected to their nearest neighbours and intensity trajectories were obtained and subsequently filtered using a Chung-Kennedy filter (*Chung and Kennedy, 1991*) to reduce background noise. Using an additional custom written algorithm, steps in the obtained intensity trajectories could be identified and were then subject to pre-defined quality controls to assess how the step-function fit to the obtained trajectories. These thresholds were namely: (i) The spot finder threshold (θLM)=500. This value was empirically set such that the number of true dimer events discarded was minimal and false positives were not incorporated into the data set. (ii) R2 value (how well the step-function fits the trajectory)

=0.95. This threshold cannot incorporate bias into the data as too high an R2 value discards single and two step intensity profiles to the same degree. (iii): θj: The maximum change in spot location following bleaching. This is set to three standard deviations above the mean. Due to the spots being well-immobilized, change in spot position was rare and minimal where occurring.

### Identification of complexes

Complexes were identified and tracked using a TrackMate plugin implemented in Image J version 2.0.0. Estimated spot diameters and an empirically determined threshold for the number of spots detected were entered for each individual cell before implementation of a tracking algorithm based on reference (*Jaqaman et al., 2008*).The resultant track schemes outputted quantitative data used in analysis.

## Molecular dynamics simulation procedure

The CHARMM22/CMAP all-atom force field (http://pubs.acs.org/doi/abs/10.1021/jp973084f) was used with explicit TIP3P waters using GROMACS 5.0.3 (*Bjelkmar et al., 2010*). Lipid A parameters compatible with CHARMM22 were used, which correctly reproduce structural and dynamic properties of lamellar phases as described (*Paramo et al., 2015*). Starting simulations were setup based on the crystal structure of the LPS-bound, heterotetrameric TLR4/MD-2 (*Park et al., 2009*) (pdb: 3FXI). The ligand-bound or ligand-free apo complexes were setup as described previously (*Paramo et al., 2013*). Briefly, each complex was placed in an octahedral unit cell (dimension ~17 nm) and solvated with a 0.1 M NaCl solution. Energy minimization using steepest descents was performed (<10,000 steps) to remove steric clashes, and a 1.5 ns position-restrained equilibration phase followed. Subsequently, 100 ns production simulations were initiated for each system in the NpT ensemble. Since no experimental structures are presently available for apo, dimeric TLR4, three production simulation replicas of this system were initiated using different initial random velocities (*Figure 2—figure supplement 2*). Equations of motion were integrated using a 2 fs time step with bond lengths constrained via LINCS (*Hess, 2008*). Lennard-Jones interactions were smoothly switched off between 1 nm and 1.2 nm, and electrostatics were computed using the Particle-Mesh-Ewald algorithm (*Essmann et al., 1995*) with a 1.2 nm real-space cutoff. Temperature and pressure were coupled using the velocity-rescale thermostat (*Bussi et al., 2007*) at 298 K and the Parrinello-Rahman barostat (*Nosé and Klein, 1983*; *Parrinello and Rahman, 1981*) at 1 atm, respectively.

## Principal Component Analysis (PCA)

PCA can be used to remove the high-frequency 'background' motions from trajectories simulation trajectories, in order to identify collective, low-amplitude protein dynamics. PCA was performed by calculating and diagonalizing the mass-weighted covariance matrix for the C-alpha atoms of each pair of TLR4 chains in the dimer. The corresponding trajectory was projected onto the first eigenvector, and interpolation between the two extreme projections around the average structure were used to generate porcupine plots within the VMD package (*Humphrey et al., 1996*).

## Acknowledgements

This work was supported by grants from the Medical Research Council (G1000133) to NJG and CEB. and a Wellcome Trust Investigator award to NJG. (WT100321/z/12/Z) and to CEB (WT108045AIA). We would like to thank GSK for the gift of CRX555, Iain Fraiser (NIH) for the reporter macrophages, Doug Golenbock and Kate Fitzgerald (University of Massachusetts Medical School) for a gift of the immortalized wild type, MyD88$^{-/-}$ and TLR4$^{-/-}$ macrophage cell lines (commercially available from BEI; see Materials and methods). Data from this paper are archived at http://dx.doi.org/10.17863/CAM.6018.

## Additional information

### Funding

| Funder | Grant reference number | Author |
|---|---|---|
| Medical Research Council | G1000133 | Nicholas J Gay<br>Clare E Bryant |
| Wellcome Trust | WT100321/z/12/Z | Nicholas J Gay |
| Wellcome Trust | WT108045AIA | Clare E Bryant |

The funders had no role in study design, data collection and interpretation, or the decision to submit the work for publication.

### Author contributions

Sarah Louise Latty, Lee Hopkins, Data curation, Formal analysis, Investigation; Jiro Sakai, Data curation, Formal analysis, Investigation, Visualization; Brett Verstak, Resources, Validation, Methodology; Teresa Paramo, Software, Formal analysis; Nils A Berglund, Data curation, Software, Formal analysis, Validation; Eugenia Cammarota, Pietro Cicuta, Analysed data and wrote the software for data analysis; Nicholas J Gay, Conceptualization, Funding acquisition, Writing—original draft, Writing—review and editing; Peter J Bond, Software, Formal analysis, Methodology; David Klenerman, Clare E Bryant, Conceptualization, Supervision, Funding acquisition, Investigation, Methodology, Writing—original draft, Project administration, Writing—review and editing

### Author ORCIDs

Jiro Sakai (iD) http://orcid.org/0000-0002-2526-2766
Nicholas J Gay (iD) https://orcid.org/0000-0002-2782-7169
David Klenerman (iD) https://orcid.org/0000-0001-7116-6954
Clare E Bryant (iD) https://orcid.org/0000-0002-2924-0038

### Decision letter and Author response

Decision letter https://doi.org/10.7554/eLife.31377.020
Author response https://doi.org/10.7554/eLife.31377.021

## Additional files

### Supplementary files

• Supplementary file 1. Structural properties of TLR4/MD2 heterotetrameric complex observed during final 20 ns of molecular dynamics simulations The modeled TLR4/MD-2 heterotetramer exhibited increased structural drift with respect to the LPS-bound X-ray structure in the absence of ligand (apo state), as reflected in the mean RMSD values. Measurement of the surface areas buried between protein chains reveals that the largest conformational changes are evident at the primary TLR4 dimerization interfaces (which govern the stability of the higher-order heterotetrameric complex) than at the secondary TLR4 dimerization interfaces, with shifts of up to ~ 60% versus~20%, respectively. This is consistent with the observed relative motion of up to ~ 10 Å of MD2 relative to its primary TLR4 partner in the apo state, as described in the main text.
DOI: https://doi.org/10.7554/eLife.31377.015

• Transparent reporting form
DOI: https://doi.org/10.7554/eLife.31377.016

### Major datasets

The following previously published dataset was used:

| Author(s) | Year | Dataset title | Dataset URL | Database, license, and accessibility information |
|---|---|---|---|---|
| Latty S, Sakai J, | 2016 | Research data supporting | https://doi.org/10.17863/ | Available at Apollo - |

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
