## [Decision Letter]

Thank you for submitting your article "Activation of Toll-like receptors nucleates assembly of the MyDDosome signaling hub" for consideration by *eLife*. Your article has been favorably evaluated by Tadatsugu Taniguchi (Senior Editor) and three reviewers, one of whom, Michael L Dustin (Reviewer #1), is a member of our Board of Reviewing Editors.

The reviewers have discussed the reviews with one another and the Reviewing Editor has drafted this decision to help you prepare a revised submission.

We have taken note of the overlapping effort that was published on November 1. However, our policy is to protect those articles submitted to *eLife* prior to the publication of a competing effort. Thus, we ask you to proceed with the essential revisions described below with no concern about the other now published work. Of course, it would be appropriate to acknowledge this work in your revised submission.

Summary:

There is a consensus among reviewers that this is an innovative approach with potential to advance understanding of TLR signaling. The manuscript uses live-cell imaging to examine the dynamic behavior of TLR signaling components and thus establishes an important approach for advancing the field. Experiments demonstrate dynamic changes in TLR4 dimerization status and make a convincing case against ligand-induced TLR4 oligomerization at the membrane. The manuscript nicely assembles a time-resolved sequence of signaling events. However, there are a few concerns with interpretation of the data and key conclusions.

Essential revisions:

1) Can the authors provide evidence that the putative MyDDosomes are functional in terms of down-stream signaling? This would also help resolve if the MyD88-GFP is able to participate in functional complexes. Along these lines a general issue is that single cell analysis typically requires well-separated molecules, which may be a challenge at physiological densities of a surface receptor. The authors should be lauded for extensive use of bona fide macrophages from knockout mice that are rescued to express the tagged receptors. But does the level of expression of the MyD88 and TLR4 match physiological expression levels that might be seen in the reporter line used for Figure 3. The authors validate the labeled constructs in 293T cells, but they don't show that the labeled receptor rescue signaling in the macrophages.

2) Can the authors provide more information about the kinetics of TLR dimer formation? There is some concern that only a 2-fold increase is seen with 10 nM LPS at 5 minutes and higher concentrations of LPS seem to accomplish TLR4 dimerization before it can be investigated with these time points. Can data from a shorter time point be provided to better understand how TLR dimerization would be related to initiation of MyDDosome formation?

3) A better explanation of how the P712H mutation and how it leads to conclusions in the last paragraph of the subsection “TLR4 agonists induce formation of receptor dimers but not higher order oligomers”, is requested.

---

## [Author Response]

Essential revisions:1) Can the authors provide evidence that the putative MyDDosomes are functional in terms of down-stream signaling? This would also help resolve if the MyD88-GFP is able to participate in functional complexes. Along these lines a general issue is that single cell analysis typically requires well-separated molecules, which may be a challenge at physiological densities of a surface receptor. The authors should be lauded for extensive use of bona fide macrophages from knockout mice that are rescued to express the tagged receptors. But does the level of expression of the MyD88 and TLR4 match physiological expression levels that might be seen in the reporter line used for Figure 3. The authors validate the labeled constructs in 293T cells, but they don't show that the labeled receptor rescue signaling in the macrophages.

This point is something that we considered very carefully during the preparation of our manuscript. TLR4 and MyD88 expression varies quite widely depending on cell type and conditions so it is not clear what physiological levels are, also TLR4 trafficking is regulated so the total expression and the amount of receptor at the surface are not correlated. Our experiments are internally controlled as TLR4 signalling is tightly coupled i.e. no MyDDosomes are seen in unstimulated cells and reduced dimers are seen after LPS treatment (Figure 2B(ii) – shows activation is coupled to endocytosis).

Expression levels of TLR4 and MyD88 in transduced cells

The problem with attempting to compare the levels of TLR4 or MyD88 in transduced cells with those present in wild-type cells is that viral transduction does not result in 100% of the cells being transduced with these constructs. We usually observe a transduction level of 40%, for example, with TLR4-Halo. Using a direct comparison of protein levels with western blot analysis or FACs analysis will, therefore, result in a lower protein level in the transduced cells because, unlike wild type cells, less than 100% of the cells will be expressing these proteins. For TLR4 we can calculate an approximate level of expression from our single cell analysis and compare it to published data (see below). We have also done the same for MyD88, but we have no reference point with which compare our calculated value of the level of MyD88 proteins so we have not added this to the text.

At a single cell level we observe approximately 30-50 TLR4 tracks per cell, over an average area of 100 µm^2^. Not all the TLR4-Halo molecules are labelled (we have previously measured Halo tag labelling efficiency as 33% in HEK cells (Biophysical Journal, 2015, 109, 1798-1806) by using antibodies). We could not, however, find a suitable and validated antibody to label only TLR4 in order to do the same experiment to measure the labelling efficiency directly. Assuming a similar efficiency value of 33%, then the estimated number of TLR4 is approximately1per µm^2^. This is close to the levels used for a recently published paper (Sci. Signal. 31 Oct 2017, Vol. 10, Issue 503, eaan1308) less than 4 per µm^2^, where TLR4 has been transfected into HEKs at levels comparable to endogenous TLR4 in glioma cells. In this paper the authors present similar data to ours with respect to TLR4 dimers (preformed dimers were present, the fraction increased on adding LPS and no oligomers were seen). We have added the following text to the methods section:

“In single cell imaging we observe approximately 30-50 TLR4 tracks per cell, over an average area of 100 µM^2^. […] This is similar to the levels of TLR4 described when transfected into HEKs at levels considered comparable to endogenous TLR4 in glioma cells (Sci. Signal. 31 Oct 2017, Vol. 10, Issue 503, eaan1308).”

Labelled molecules are functional

TLR4-Halo and MyD88-GFP both signalled to NF-κB in HEKs therefore the chances of these constructs not signalling in macrophages is unlikely. We performed a number of studies with constructs tagged with a range of fluorophores and MyD88 only signalled with a GFP tag, other tags prevented signalling. To address this point, however, we have virally transduced TLR4-Halo into the TLR4^-/-^cells and shown in single cells that this construct signals by stimulating the cells with LPS and immunolabelling NFkB-p65 translocation to the nucleus (Figure 2—figure supplement 1). Likewise we have transduced MyD88-GFP into MyD88^-/-^ immortalised macrophages to show signalling occurs in macrophages transduced with MyD88-GFP (Figure 1—figure supplement 1). These experiments show that the labelled constructs used in our experiments are functional in macrophages and that they are also expressed at a sufficient level to allow signalling in response to cellular stimulation. We have added these figures and legends to Figure 1—figure supplements 1 and 2).

2) Can the authors provide more information about the kinetics of TLR dimer formation? There is some concern that only a 2-fold increase is seen with 10 nM LPS at 5 minutes and higher concentrations of LPS seem to accomplish TLR4 dimerization before it can be investigated with these time points. Can data from a shorter time point be provided to better understand how TLR dimerization would be related to initiation of MyDDosome formation?

There is an inherent dead time in our experiments of a few minutes. This is due to the time taken for the cells to adhere to the cover slides preceding imaging. Alignment of the instrument for TIRF imaging also prevents us from probing earlier time points. We have added the following text to the manuscript:

“There is an inherent dead time of a few minutes due to the time taken for the cells to adhere to the cover slides preceding imaging”

3) A better explanation of how the P712H mutation and how it leads to conclusions in the last paragraph of the subsection “TLR4 agonists induce formation of receptor dimers but not higher order oligomers”, is requested.

These sentences have been replaced with:

“This result indicates that LPS binding to the Pro712His receptor stabilises the TLR4/MD-2 heterodimer and inhibits the formation of active tetramers. Dimerisation of the TLR4/MD-2 ectodomain and the TIR domain BB-loop are thus synergistic and both events must occur to produce an active receptor complex. This implies a two-step activation mechanism for TLR4, as illustrated in Figure 2C.”